# Acute Effect of a Half-Marathon over the Muscular Function and Electromyographic Activity of the Pelvic Floor in Female Runners with or without Urinary Incontinence: A Pilot Study

**DOI:** 10.3390/ijerph20085535

**Published:** 2023-04-17

**Authors:** Horianna Cristina Silva de Mendonça, Caroline Wanderley Souto Ferreira, Alberto Galvão de Moura Filho, Pedro Vanderlei de Sousa Melo, Ana Flávia Medeiros Ribeiro, Kryslly Danielle de Amorim Cabral, Renato de Souza Melo, Leila Maria Alvares Barbosa, Ana Paula de Lima Ferreira

**Affiliations:** Department of Physical Therapy, Federal University of Pernambuco (UFPE), Recife 50740-521, Brazilana.lferreira@ufpe.br (A.P.d.L.F.)

**Keywords:** running, pelvic diaphragm, electromyography, urinary incontinence

## Abstract

Objective: to verify the acute effect of running a half marathon on pelvic floor muscle (PFM) function and electromyographic (EMG) activity in female runners with and without urinary incontinence. Methods: This is a cross-sectional pilot study. The sample was divided into two groups: runners with urinary incontinence (with UI) and runners without urinary incontinence (without UI). A semi-structured form and the International Consultation on Incontinence Questionnaire—Short Form (ICIQ-UI-SF) were used for data collection. The EMG and PFM function were evaluated using the PERFECT method before and immediately after running a half marathon. Results: A total of 14 runners were included (8 with UI; 6 without UI). Runners with and without UI did not show significant differences for EMG and PERFECT. The acute effects of the half marathon on runners without UI were reduced PFM function in terms of strength (*p* = 0.00), reduced endurance (*p* = 0.02), and reduced repetition (*p* = 0.03), and an increase in EMG measured by the median frequency (*p* = 0.02). Runners with UI showed reduced PFM function in terms of strength (*p* = 0.05) and repetition (*p* = 0.01). Conclusion: there was no difference in the acute effects of the half marathon on PFM function and EMG in women with and without UI.

## 1. Introduction

The pelvic floor (PF) is formed of muscles, fascia and ligaments whose main functions are to support the pelvic organs, continence and sexual function. The active support of the strength and coordination of pelvic floor muscles (PFMs) associated with passive support (ligaments and fascia) is an essential factor for the conscious and unconscious performance of PF muscle recruitment [1,2,3].

The practice of high-impact sports can influence PFM function and cause PF dysfunction such as urinary incontinence (UI), which is one of the most common dysfunctions [4]. The International Continence Society (ICS) defines UI as any involuntary loss of urine, being classified as stress urinary incontinence (SUI), urge urinary incontinence (UUI) and mixed urinary incontinence (MUI). SUI is the most recurrent type as a result of physical exercise, coughing or sneezing [5,6]. The prevalence of UI in athletes involved in running sports is 44% [7]. However, the prevalence of UI associated with different running distances is not known, and the prevalence of UI in female half-marathon runners has not been studied [6,7].

Street running is considered a low-cost and easily accessible sport, and ranks first among the physical activities practiced by the world’s population, with the growing participation of females [8]. Women half-marathon runners increase the time, volume and frequency of their exercise/training to improve their performance. This may contribute to failures in PFM contraction and in relaxation mechanisms and intra-abdominal pressure (IAP) control [2,9,10]. Single-leg impact on the ground, which characterizes running, causes PFM overload [9]. In addition, the ground reaction force exerted on PFMs during running also increases the IAP. These events can be decisive in the development of UI [11,12].

Electromyography (EMG) has proven to be a reliable and widely used tool to verify PFM recruitment in female runners [13]. Although the PERFECT method is a tool which is considered to be moderately reliable for verifying PF strength in scientific research, it has also been widely used in clinical practice to evaluate PFM function [14].

EMG has been used more than PERFECT to analyze the effects of running on the PFMs in women with and without UI. In these studies, different speeds (7.11 and 15 km) performed on a treadmill for periods of time were compared, with it being necessary to perform between 6 and 70 steps [2,13,15,16]. Thus, greater pre-activation of the PFMs in women without UI was observed at speeds of 11 km/h in these studies. However, the authors suggested that EMG is not only analyzed at different speeds, but with different distances covered. Above all, the importance of analyzing EMG in long-distance running has been emphasized, since the successive and uninterrupted impact on the PFMs at longer distances may be more relevant to alterations appearing in EMG activity [13,17,18].

Given the above, this study aimed to analyze the acute effect of a long-distance race (half marathon) on the function and electromyographic activity of the pelvic floor muscles of female runners with and without urinary incontinence.

## 2. Material and Methods

### 2.1. Type of Study and Eligibility Criteria

This is a cross-sectional pilot study consisting of female adult half-marathon runners divided into 2 groups: those with urinary incontinence (with UI) and those without urinary incontinence (without UI).

The study was conducted in the Laboratory of Kinesiotherapy and Manual Therapeutic Resources (LACIRTEM), in the Laboratory of Physiotherapy for Women’s Health and Pelvic Floor (LAFISMA) located in the Department of Physiotherapy at the Federal University of Pernambuco (UFPE), and in three private Physiotherapy Clinics located in Pernambuco. The participants were recruited through publicity on social networks and visits to running groups in the city of Recife, PE, Brazil, and the metropolitan region.

All participants were informed about the research stages and signed the informed consent form (ICF). This study was approved by the Research Ethics Committee of the Federal University of Pernambuco (CAEE: 41984920.2.0000.5208; Opinion: 4.524.121) which deals with the Code of Ethics for Research with Human Beings.

The following inclusion criteria were established: female half-marathon distance runners (21 km), aged between 18 and 50 years, who had been practicing road running for at least 1 year.

The exclusion criteria were women with diseases that could be associated with the onset of UI or neuromuscular disease, with a history of miscarriage/abortion dating back to less than 1 year ago, with a history of previous urogynecological/abdominal surgery, who showed the presence of urinary and/or vaginal infection, using the pilates method as a complement to physical training, who were undergoing physiotherapeutic treatment for UI, who had never had sexual intercourse, and who showed the presence of musculoskeletal injuries, which would have reduced the runner’s performance during the participation period in the study [19]. Women in the climacteric phase were also excluded.

### 2.2. Evaluation Criteria

The runners initially answered questions about their sociodemographic and anthropometric data (age, weight, height, body mass index and education), in addition to urinary, gynecological, and obstetrical symptoms, as well as training regimen.

Then, the International Consultation on Incontinence Questionnaire—Short Form (ICIQ-UI-SF) was applied to investigate the presence, frequency, and severity of urine loss and its impact on their quality of life. The ICIQ-UI-SF score ranges from 0 to 21 points and the higher the score, the worse the impact of UI on quality of life. The ICIQ-UI-SF was also used as a criterion to define whether or not the volunteers had urinary incontinence. Women who answered “Yes” to question number 6 of the ICIQ-UI-SF were considered incontinent [20].

Physical assessments (EMG and PERFECT) were performed before and immediately after the half-marathon at the starting and finishing locations of the race. The runner was instructed to remain in a supine position with her head resting on a pillow, hips flexed and abducted, and knees flexed. The external genitalia were examined in this posture to identify the presence or absence of a visible voluntary contraction of the PFMs after a verbal command for the maximum contraction time [21,22].

A physical examination was performed using the PERFECT method for the functional assessment of PFMs. The subject was instructed to lie down in the dorsal decubitus position with her head resting on a pillow and with flexed hips and abducted and flexed knees. First, the external genitalia were observed to identify whether there was a visible voluntary contraction of the PFMs after the verbal command. Then, the researcher/examiner used procedure gloves and lubricating gel and performed bidigital introduction into the vagina. The parameters evaluated were strength on the Oxford scale (strength), resistance in seconds (endurance), number of sustained contraction repetitions (repetition) and number of fast PFM contraction repetitions [23].

A Miotec^®^ surface electromyograph at a 20 Hz high pass and 500 Hz low pass with 60 Hz notch filters was used to assess PFM recruitment. Disposable surface electrodes (disposable electrodes for children by the Meditrace^®^ brand) with low impedance made with a hypoallergenic medical adhesive with high adherence, double contact Ag/AgCl (Silver/Silver Chloride) and hydrogel were used to capture the electromyographic signal. The volunteers were instructed to empty their bladder if it was full before performing this exam [21].

Next, the surface electrodes were positioned at the three and nine o’clock positions in the perianal region. The electrodes were fixed in these regions in order to acquire the electromyographic activity of the pelvic floor. A reference electrode on the right anterior superior iliac spine was used in order to eliminate external interference [21].

The electromyographic activity at rest and after two minutes of rest was initially recorded, and then the volunteer was asked to perform 3 maximum voluntary contractions (MVCs) of the PFMs. The three MVCs were sustained for three seconds, with a 30 s interval between each contraction [21].

After passing the filter, a contraction period of 500 milliseconds (ms) of the second MVC was selected. The root mean square (RMS), peak rms and median frequency values were recorded [21]. The values related to the signal amplitude (RMS) were normalized with the baseline, from which a contraction period of 500 ms was selected during rest that preceded the second contraction [21].

### 2.3. Half-Marathon Run

The participants were instructed to run at their usual pace (the average pace of the race, measured in minutes per kilometer) on asphalt and on a course without slopes or hills. The distance covered for a half-marathon is 21 km. Depending on the convenience of each volunteer, the chosen route could be through University campus or on one of the two shores of the city of Recife or Olinda. The runners were instructed to wear the gym clothes and sneakers they frequently used during their training sessions. The athletes were hydrated during the course according to individual needs.

### 2.4. Data Analysis

Data were tabulated in Excel Microsoft 365 and analyzed using version 20.0 of the Statistical Package for Social Sciences (SPSS) software program. The Kolmogorov–Smirnov test was used to identify data normality. Descriptive statistical analysis was performed using the mean, standard deviation and confidence intervals for continuous quantitative variables and using percentages for dichotomous variables.

Student’s *t*-test was used in cases of normality and the Mann–Whitney test was used for cases of non-normality to compare the sociodemographic and anthropometric variables and the training regimes of women with and without UI. A comparison of the dichotomous variables of comorbidities, urinary symptoms and gynecological-obstetric characteristics between the women with and without UI was performed using Fisher’s exact test.

The analysis of comparisons of PFM function and electromyographic activity between women with and without UI was performed using the Mann–Whitney test. The Wilcoxon test was used for the analysis of women with and without UI before and immediately after the race. The significance level of the tests was *p* ≤ 0.05.

## 3. Results

In total, 14 women half-marathon runners, 8 women with UI (57.1%) and 6 women without UI (42.9%) participated. Regarding the types of UI, 6 women (75%) had SUI, 1 (12.5%) had UUI, and 1 (12.5%) had MUI. Moreover, zero UI severity was observed for 4 women (50%), mild severity was observed for 3 women (37.5%), and moderate severity was observed for 1 woman (12.5%). No difference was observed regarding the sociodemographic and anthropometric characteristics or the running training regimes of women runners with and without UI (Table 1).

The mean ICIQ-SF score for women with UI was 3.63 on a scale of 0 to 21 points. It was observed that 3 (21.4%) women with UI reported losing urine during training and 7 women (87.5%) had to interrupt training to urinate (Table 2).

Muscle recruitment and the PFM functional assessment performed before and immediately after the half-marathon race are shown in Table 3, showing that there was no difference between runners with and without UI. The intragroup results demonstrate that the acute effects of the half-marathon in women without UI were a reduction in PFM function in terms of strength (*p* = 0.00), a reduction in endurance (*p* = 0.02), and a reduction in repetition (*p* = 0.03). There was also a significant increase in PFM recruitment measured by the median frequency of the EMG activity (*p* = 0.02). The acute effects of the half-marathon on PFM function for women with UI were reduced strength (*p* = 0.05) and repetition (*p* = 0.01).

## 4. Discussion

The main findings of this study reveal that the two groups of female half-marathon runners (with and without UI) were homogeneous in terms of sociodemographic and anthropometric characteristics and training regimes. Furthermore, there was no difference between the groups regarding muscle recruitment, as verified through electromyography and the PFM functional assessment through the PERFECT method before and after the race. However, a decrease was verified in both groups when comparing the data before and immediately after the half marathon in the strength and number of repetitions of sustained contractions. Moreover, a decrease in endurance and an increase in the median frequency of electromyography was only observed in the group of runners without UI.

The prevalence of incontinent runners present in this study was 57.1%, with the most frequent type being SUI (75%). The high UI prevalence is directly related to the practice of physical activity and high-impact sports such as running [4,7]. Studies performed by Pires et al. (2020) pointed out the 44% prevalence of UI in female runners, but the occurrence of SUI was not reported. Studies on SUI are scarce in female long-distance runners; however, the prevalence of SUI is generally high among female athletes. Rodriguez et al. (2022) found prevalence of SUI in 64.4% of the female track and field athletes, and Velazquez et al. (2021) found a prevalence of SUI of 89.3% [7,24,25].

The high percentage of women who reported having to interrupt training to urinate (87.5%) may have been related to the intensity and duration of the race, body movements and the impact of the ground reaction force, which can vary between 2.4 to 3.9 times the body weight over the PFMs, favoring urinary loss. It is worth mentioning that the athletes in a half-marathon perform two straight hours of activity with continuous impact on the PFMs [2,10].

The PFM function and electromyographic activity observed in this study did not present significant differences when comparing female runners with and without UI. These findings differ from those of other studies that showed that women with UI had lower PFM function and EMG activity than female runners without UI did [18]. These results may be related to the small sample size in each group of female half-marathon runners who participated in the present study.

The reduction in the strength and number of sustained PFM contractions observed in women with UI in this study is in line with that observed in the studies by Araújo et al. (2015); Pires et al. (2020); and De Melo et al. (2020). These authors justified these effects due to the impact of running promoting an increase in IAP and negatively influencing PF structures. These factors increase overload and tend to weaken the PFMs [1,4,10].

Similarly to women with UI, women without UI also showed a reduction in the strength and number of sustained contractions, suggesting that PFM overload alone may not be a determining factor for the occurrence of urinary incontinence in female half-marathon runners. PFM strength and endurance alone may not have the potential to determine the presence of UI in female runners [4,8,26]. However, the failure of the PFMs and abdominal muscles to co-contract may contribute to the onset of UI in these women [4,26,27,28].

The literature points to two opposing hypotheses in which running can strengthen or overload the PFM of female runners [3,4]. The first hypothesis proposes that failure in the co-contraction of the PFM and abdominals may occur with the increase in IAP resulting from running impact. In this case, runners become unable to maintain the contraction force for a prolonged period, causing overload, chronic injuries, PFM weakness and UI [3,4,29]. The second hypothesis proposes that there is a simultaneous reflex co-contraction of the abdominal muscles and/or pre-contraction of the PFM under running impact. These events contribute to better muscle conditioning due to the increased recruitment of these muscles and consequent improvement in continence mechanisms [3,4,27,28].

When the co-activation mechanism between the PFMs and abdominal muscles is altered, an increase in PF dysfunctions and the development of UI may occur [27,28,30]. The present study aimed to analyze the acute effects of a half-marathon on the muscle function and electromyographic activity of the pelvic floor, as we believe that these repeated efforts over months and years of training and competitions may contribute to more significant functional changes, especially in runners who develop failures when pre-contracting the pelvic floor muscles and abdominal muscles for long activity periods. This question is being explored in studies which continously being developed by our research group. In doing so, we hope to clarify this gap.

In the present study, women without UI showed greater electromyographic activity after performing the half-marathon, while women with UI did not show a significant difference for this parameter. These findings were also observed by Koenig et al. (2020), and may be related to the fact that women without UI may have better co-activation between the abdominal and PF muscles [16,29].

This study has some limitations such as the small sample size, which was due to the difficulty in accepting the athletes to be part of the study. This difficulty is justified by the discomfort involved in exposing intimate parts of the body, which was necessary during the physical assessment, as well as the use of a questionnaire with intimate questions about sexual activity and symptoms related to the PF region. Conducting the study during the COVID-19 pandemic period also made it difficult for the athletes to participate.

To our knowledge, this study is the first to investigate electromyographic function and activity in female half-marathon runners. Therefore, it contributes to filling this gap and to the self-knowledge and education of these runners regarding the prevention and treatment of UI. Lectures and the distribution of booklets at the end of the study were not only important for prevention, but also for paradigm shifts in coping with UI symptoms and for referral to specialized physical therapy treatments.

In addition, this study disseminated useful information about UI for physical education professionals, physicians, physiotherapists and other professionals in the field of sports, as well as to women runners, with an extension to the community where these women live. Furthermore, it was possible to encourage sports professionals to discuss and be committed to identifying signs and symptoms of UI in female runners. Thus, it is possible to prevent and/or refer them to physiotherapeutic treatment when necessary in order to improve their quality of life and sports performance.

## 5. Conclusions

Adult female half-marathon runners, with and without UI, assessed by electromyography and the PERFECT method, did not show any differences in their PFM function.

Our research group is continuing investigations in order to increase the sample size and clarify some questions regarding the chronic effect and influence of some outcomes, such as failures in the pre-contraction of the pelvic floor muscles and abdominal muscles for long periods of running. These questions are being explored in studies that continue to be developed by our research group with the aim of providing more evidence on the subject.

## Figures and Tables

**Table 1 ijerph-20-05535-t001:** Sociodemographic and anthropometric characteristics, as well as the training regimen of women half-marathon runners with and without urinary incontinence (n = 14). Pernambuco, Brazil, 2022.

	Runners with UI (n = 8)	Runners without UI (n = 6)	
	Mean ± SD CI (95%)	Mean ± SD CI (95%)	*p*-Value
Age (years)	38.75 ± 2.56 (32.68–44.82)	34.43 ± 2.41 (28.51–40.35)	0.24
Weight (kg)	61.37 ± 2.54 (55.34–67.40)	66.00 ± 3.65 (57.07–74.93)	0.30
Height (cm)	1.63 ± 0.01 (1.59–1.67)	1.65 ± 0.27 (1.58–1.71)	0.56
BMI (kg/m^2^)	23.13 ± 0.83 (21.08–25.19)	23.43 ± 1.44 (19.89–26.96)	0.85
Education (years)	14.63 ± 0.37 (13.74–15.51)	13.29 ± 0.60 (11.78–15.22)	0.07
Time running (years)	11.63 ± 3.25 (3.92–19.33)	5.00 ± 1.04 (2.44–7.56)	0.91
Running frequency (weekly)	3.25 ± 0.36 (2.38–4.12)	3.43 ± 0.36 (2.53–4.33)	0.73
Distance run (km/week)	33.50 ± 6.45 (18.33–48.77)	34.00 ± 5.94 (19.46–48.54)	0.95
Time running (min/day)	67.88 ± 4.03 (58.33–77.42)	68.57 ± 8.84 (40.39–89.61)	0.94
Pace (min/km)	5.74 ± 0.25 (5.14–6.33)	5.75 ± 0.18 (5.27–6.24)	0.07

Legend: UI = urinary incontinence; SD = standard deviation; CI = confidence interval; BMI = body mass index; kg = kilogram; cm = centimeters; m^2^ = meters squared; min = minutes; km = kilometers. Student’s *t*-test was performed.

**Table 2 ijerph-20-05535-t002:** Comorbidities, urinary symptoms and gynecological-obstetric characteristics of women half-marathon runners with and without urinary incontinence (n = 14). Pernambuco, Brazil, 2022.

	Runners with UI (n = 8)	Runners without UI (n = 6)	
	N (%)	N (%)	*p*-Value
Constipation			0.60
Yes	4 (50%)	2 (33.4%)	
No	4 (50%)	4 (66.6%)	
Nocturia			0.62
Yes	3 (37.5%)	2 (33.4%)	
No	5 (62.5%)	4 (66.6%)	
Loss of urine while running			0.20
Yes	3 (37.5%)	0 (0.0%)	
No	5 (62.5%)	6 (100%)	
Interrupt running to urinate			0.04
Yes	7 (87.5%)	2 (33.4%)	
No	1 (12.5%)	4 (66.6%)	
Pain during sexual intercourse			0.46
Yes	2 (25%)	0 (0.0%)	
No	6 (75%)	6 (100%)	
Gestation			0.59
Yes	4 (50%)	4 (66.6%)	
No	4 (50%)	2 (33.4%)	
Parity			0.61
Yes	5 (62.5%)	2 (33.4%)	
No	3 (37.5%)	4 (66.6%)	
Miscarriage/abortion			0.66
Yes	2 (25%)	2 (33.4%)	
No	6 (75%)	4 (66.6%)	

Legend: UI: urinary incontinence. Fisher’s exact test was performed.

**Table 3 ijerph-20-05535-t003:** Comparison of mean function (PERFECT method) and PFM electromyographic activity within and between groups of female half-marathon runners with and without UI. Pernambuco, Brazil, 2022.

	Runners with UI (n = 8)	Runners without UI (n = 6)	
	Mean ± SD CI (95%)	Mean ± SD CI (95%)	** *p*-Value
Strength (on the Oxford scale)			
Before the race	3.00 ± 0.26 (2.37–3.63)	3.00 ± 0.36 (2.06–3.94)	
After the race	2.38 ± 0.26 (1.75–3.00)	2.00 ± 0.36 (1.06–2.94)	0.35
*** *p*-value**	0.05	0.00	
Endurance (strength maintenance in seconds)			
Before the race	3.63 ± 0.37 (2.74–4.51)	4.00 ± 0.68 (2.24–5.76)	
After the race	3.00 ± 0.26 (2.37–3.63)	3.00 ± 0.51 (1.67–4.33)	0.54
*** *p*-value**	0.09	0.02	
**Repetition** (number of repetitions of sustained contractions)			
Before the race	4.25 ± 0.41 (3.28–5.22)	3.67 ± 0.33 (2.81–4.52)	
After the race	3.50 ± 0.32 (2.73–4.27)	2.67 ± 0.33 (1.81–3.52)	0.60
*** *p*-value**	0.01	0.03	
Fast (number of fast contractions)			
Before the race	5.00 ± 0.56 (3.66–6.34)	5.50 ± 0.67 (3.78–7.22)	
After the race	4.50 ± 0.65 (2.95–6.05)	4.33 ± 0.76 (2.38–6.29)	0.68
*** *p*-value**	0.48	0.20	
Peak RMS (µV)			
Before the race	40.25 ± 7.35 (22.85–57.65)	32.33 ± 8.18 (11.29–53.38)	
After the race	46.38 ± 5.10 (34.30–58.45)	38.17 ± 6.22 (22.16–54.17)	0.30
*** *p*-value**	0.36	0.52	
Mean RMS (µV)			
Before the race	36.50 ± 6.20 (21.82–51.18)	29.83 ± 7.52 (10.50–49.17)	
After the race	40.38 ± 5.04 (28.44–52.31)	34.00 ± 5.80 (19.08–48.92)	0.47
*** *p*-value**	0.83	0.59	
Median frequency (Hz)			
Before the race	112.88 ± 9.38 (90.68–135.07)	93.50 ± 8.91 (70.60–116.40)	
After the race	112.63 ± 7.65 (94.53–130.72)	121.33 ± 9.18 (97.72–144.95)	0.51
*** *p*-value**	0.77	0.02	

Legend: UI = urinary incontinence; SD = standard deviation; RMS = root mean square; µV = microvolts; Hz = hertz. * Intragroup *p*-value of women with and without UI (before and after the half marathon). ** Intergroup *p*-value of women with and without UI (after completing the half marathon). TheMann–Whitney test was performed.

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
