# Peer review of "Acute Effect of a Half-Marathon over the Muscular Function and Electromyographic Activity of the Pelvic Floor in Female Runners with or without Urinary Incontinence: A Pilot Study"

_ijerph, 2023, doi:10.3390/ijerph20085535_

Round 1

Reviewer 1 Report

The article is good and innovative, although the small sample size makes it necessary to be cautious in its interpretation. There are also some aspects that need to be corrected:

- In terms of methodology, the PERFECT method is not explained. L111.

- It is important to identify those results obtained through non-parametric tests as their results need to be interpreted with caution.

- I think it is important to take into account the variable time running practice (years) as there are big differences in the two groups. In addition, the standard deviations are very high.

- I do not think the interpretation of the prevalence of women with UI (L 199) is very accurate. It is a percentage of only 14 people selected and who agreed to take part in the study.

- The format of quotations is not in line with the style of the journal L219

Author Response

- We have rewritten the PERFECT exam in detail in paragraph 4 (Evaluation criteria)

- The information about the yests performed was described in the caption of each table

- In fact, womwn with UI had a longer practice of running, but this difference was not statistically significant, possibly due to the sample size.

- Yes, we agree with that placement. The difficulty in managing to increase the sample number is justified by the fact that data collection involves pelvic floor assesments, requiting exposure of intimate parts before and immediately after the 21 km run. This fact reduced the adherence of a larger number of volunteers.

- The citations ae correct, with numbers in square brackets

Reviewer 2 Report

Thank you  for the opportunity to review the manuscript. Although the topic is original and adds to our understanding of the impact of high-impact exercise on urinary incontinence the paper needs extensive review from an english native speaker. It was difficult to follow the paper and too many gramatical errors were present for the reviewer to concentrate in content. My suggestion is to send back the paper for a second review once the english is appropiately addressed. 

Author Response

The text has been completely redone with translation with translation performed by a native english speaker

Reviewer 3 Report

First of all, I would like to congratulate the authors for their work, since the theme seems very relevant to me. However, there are some issues that I think can be improved:

  • The training of each participant is not taken into account, beyond Pilates and I think it is significant because it can establish differences in the result that is measured
  • The sample is very small 
  • It is clear that the force after an impact exercise will be reduced, but how long is this effect maintained? is it significant for functionality? 
  • I miss the measurement of other variables that can support the conclusions

Author Response

-The Pilates was na exclusion criterion, although other types of training were not mentioned as exclusion criteria, other types of training performed by the volunteers that could interfere with the results of this research were not mentioned

- Yes, we agree with that placement. The difficulty in managing to increase the sample number is justified by the fact yhat data collection involves pelvic floor assessments, requering exposure of intimate parts before and immediately after the 21 km run. The fact reduced the adherence of a larger number of volunteers

- the study aimed to analyze the acute effects of the half marathon on muscle function and electromyographic activity of the pelvic floor, we believe that these repeated efforts over months and years of training and competitions may contribute to more significant functional changes. Especially in runners who fail to pre contract the pelvic floor muscles and abdon=minal muscles for long periods of activity (this information was added in paragraph 8 of the discussion). This question is being explored in studies that continue to be developed by our research group. In this way, we hope to clarify this gap.

- the conclusion has been rewritten

Reviewer 4 Report

The paper is interesting and takes into consideration the relationship between urinary disorders and long distance running. Despite the small sample size, the results obtained are interesting and can be used to further develop the research in the future. I suggest discussing more how this paper could be part of the increased dissemination of information in non-medical fields (e.g., physical education professionals, physiotherapists, women runners, and the communities). 

Author Response

The information was reinforced in the last paragraph of the discussion and also in the conclusion

Round 2

Reviewer 2 Report

Thank you for reviewing the manuscript and editing the english translation. 

I would suggest changing the criteria "virgin" , to "who had never had sexual intercourse". 

Author Response

We appreciate the comments and suggestions made. In response to requests made, some paragraphs were reformulated so that the text in English was better understood. The study design was changed to a cross-sectional study. And the term "virgin" has been changed to women who have never had sexual intercourse. The first paragraph of the conclusion was rewritten

Reviewer 3 Report

A longitudinal study uses continuous or repeated measures to follow a population group over a prolonged period of time. In this study, no follow-up has been carried out (at least it is not reflected) nor is the use of different variables specified over a period of time for subsequent analysis.

Author Response

We appreciate the comments and suggestions made. In response to requests made, some paragraphs of the introduction were rewritten. The study design was changed to a cross-sectional study. The conclusion has also been reworked.
